# OFDVDnet: A Sensor Fusion Approach for Video Denoising in Fluorescence Guided Surgery

**Trevor Seets**[*1]                                                                    SEETS@WISC.EDU
**Wei Lin**[*1]                                                                          WLIN77@WISC.EDU
**Yizhou Lu**[1]                                                                         YLU289@WISC.EDU
**Christie Lin**[3]                                                              CHRISTIE.LIN@ONLUME.COM
**Adam Uselmann**[3]                                                         ADAM.USELMANN@ONLUME.COM
**Andreas Velten**[1,2]                                                                  VELTEN@WISC.EDU

[1] *University of Wisconsin - Madison, Department of Electrical and Computer Engineering*
[2] *University of Wisconsin - Madison, Department of Biostatistics and Medical Informatics*
[3] *OnLume Surgical*

**Editors:** Accepted for publication at MIDL 2023

## Abstract

Many applications in machine vision and medical imaging require the capture of images from a scene with very low radiance, which may result in very noisy images and videos. An important example of such an application is the imaging of fluorescently-labeled tissue in fluorescence-guided surgery. Medical imaging systems, especially when intended to be used in surgery, are designed to operate in well-lit environments and use optical filters, time division, or other strategies that allow the simultaneous capture of low radiance fluorescence video and a well-lit visible light video of the scene. This work demonstrates video denoising can be dramatically improved by utilizing deep learning together with motion and textural cues from the noise-free video.

**Keywords:** Fluorescence guided surgery, video denoising, neural network denoising, deep learning, fluorescence video dataset, guided denoising

## 1. Introduction

In medical imaging modalities such as fluorescence image-guided surgery (FGS), thermal imaging, imaging of Cerenkov radiation in radiotherapy, or Raman scattering, there are many photon-starved or low contrast signals that are important to image at video frame rates. For example in FGS, a patient is injected with a fluorescent compound that binds to a specific tissue type. Then the tissue is illuminated with an excitation light source during surgery and the fluorescent marker emits a weak fluorescent signal that can be picked up by a fluorescent camera. Existing procedures use fluorescent markers to visualize tumors (Ishizawa et al., 2009; Zhang et al., 2017), blood vessels (Li et al., 2010), lymph nodes (Kitai et al., 2005; Frumovitz et al., 2018), necrotic tissue (Xie et al., 2015; Zajac et al., 2022), or nerves (Gibbs-Strauss et al., 2011). A surgeon will use the fluorescent camera feed to make real-time intraoperative decisions, where having high fluorescent signal is key. Fluorescence is much weaker than standard light sources and this low signal is compounded by the need for short exposure times for real-time video. Additionally, some

---

[*] Contributed equally

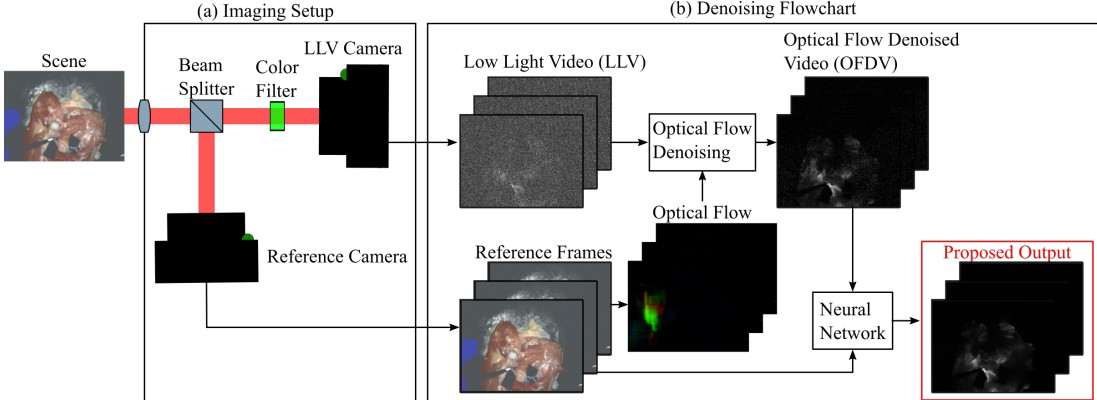

**Figure 1: Overview:** (a) Our imaging setup consists of two co-registered cameras: the reference camera measures a conventional intensity image of the scene while the low light video (LLV) camera measures a noisy fluorescence video. (b) To denoise the LLV, we first find optical flow throughout the video using the reference frames. We then apply the optical flow to the LLV to align frames in time, we merge the aligned frames to create the OFDV. The OFDV and reference frames are then fed into a video denoising neural network that will produce our final denoised video.

useful fluorescent dyes and intrinsic fluorescence compounds often have low photon yields or low specificity leading to noisy videos that may lack the required sensitivity for clinical decision-making. For example, the auto-fluorescence from the inferior parathyroid can be weak during thyroidectomy (Kim et al., 2016) making it hard to detect the target. The low signal levels in FGS can be improved computationally through video denoising methods that will be able to take into account past captured frames to denoise a current frame as well as temporal and spatial information. Denoising methods will increase signal-to-noise levels in FGS to allow surgeons to lower the injection dose, allow for longer imaging time periods, and allow use of lower contrast and low sensitivity fluorescent agents even auto-florescence.

Due to the low brightness of fluorescent markers, fluorescence signals are flooded by background and ambient illumination. Hardware solutions have been developed to increase signal by filtering out ambient light. One method, called transient lighting (Velten et al., 2020), switches on and off the room light in short intervals between the exposure of fluorescence camera frames. The blinking light is unnoticeable to the human eye due to the flicker-fusion threshold making the surgical room appear well-lit while providing a dark environment for fluorescent capture. Another method is wavelength filtering which blocks photons from the illumination light source and photons outside of the fluorescence emission band. Both wavelength filtering and transient lighting lend themselves to a two-camera approach where one camera captures the low light video (LLV) of fluorescence, and another co-located camera captures a good quality RGB video called the reference video (RV).

In this work, we aim to recover a denoised video from a noisy low light video (LLV) of a faint fluorescent compound with noise standard deviations one magnitude higher than what current methods consider. To address this extreme noise, we propose a sensor fusion approach where the RV provides motion and structural cues that act as a guide when denoising the LLV. Due to the low signal level of the fluorescent compound we consider using a shot noise-limited camera such as a single photon avalanche diode (SPAD) to capture the LLV.

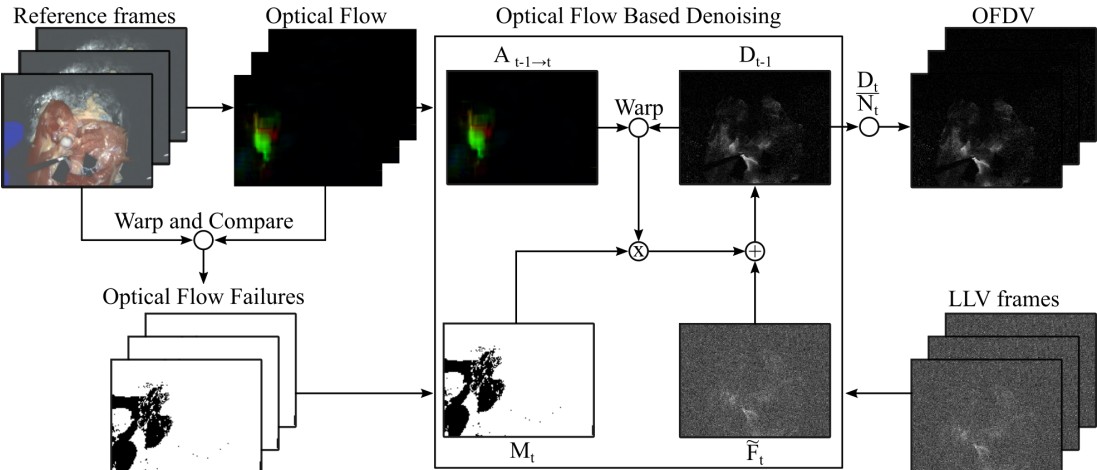

**Figure 2: Optical Flow Denoising:** First, we calculate optical flow from the RV frames. We then calculate an optical flow failure mask by warping the RV frames forward in time and comparing the warped frames to the true RV frame at that time. We then iterate over every frame and create a running sum of the LLV video. In one iteration we warp the previous running sum result forward in time 1 frame with the optical flow. We then reset pixels in the running sum according to the optical flow failure mask. The current LLV frame is then added to the result which becomes the next running sum frame and finally the OFDV.

We identify 3 useful properties for FGS denoising, spatial, temporal, and RV correlation properties, many existing algorithms were only designed to use 2 of these factors. Video denoising methods, such as the non-learning based V-BM4D (Maggioni et al., 2012) and learning based FastDVDnet (Tassano et al., 2020), DVDNet (Tassano et al., 2019), and VNLNet (Davy et al., 2019) are meant to denoise videos without a RV so use only temporal and spatial properties. These algorithms rely on explicit or implicit feature mapping to properly denoise frames, which is difficult with the extreme noise considered in this paper. Additionally, convolutional neural network (CNN) approaches do not scale well to take as input a large number of video frames due to GPU memory limitations. A second class of denoisers are guided image denoisers such as guided filtering and joint bilateral filtering that were designed for images with a guide image so make use of spatial properties and RV spatial correlations but no temporal information. A third category of denoisers are align and merge techniques (Hasinoff et al., 2016) which are often used to denoise videos generated by single photon cameras (Seets et al., 2021; Ma et al., 2020; Istvan et al., 2015; Gyöngy et al., 2017; Gyongy et al., 2018) and exploit temporal correlation; however, at extreme noise levels, alignment is difficult. To utilize all three properties, we propose OFDVDnet which draws inspiration from both align and merge techniques and deep learning methods in a two-stage approach. First, we align the LLV with motion extracted from the clean RV to compress the long temporal correlations in the LLV followed by a video denoising CNN based on FastDVDnet.

An overview of OFDVDnet is shown in Fig 1, first, our LLV and RV data is collected using two co-registered cameras. We use the RV to compute optical flow and an occlusion mask between successive frames (Fig 2). The optical flow and mask are then applied to the LLV to create a denoised motion-compensated estimate of the LLV, called the optical flow denoised video (OFDV). The OFDV incorporates information from many distant frames.

One main advantage of the OFDV is it is a compressed representation of many different frames that can be easily loaded into GPU memory. Finally, we feed the ODFV and the RV frames into a video denoising neural network inspired by FastDVDnet to create a final denoised output.

We capture a RV and LLV aligned dataset using a commercial FGS imaging system (OnLume Surgical, Madison, WI) in a simulated surgery based on the blue blood chicken model used in micro-surgical training (Albano et al., 2021) where we inject a fluorescent agent to highlight blood vasculature in chicken thighs. To collect training data we use indocyanine green (ICG) as our fluorescent agent to capture low-noise video and simulate much weaker fluorescence giving us noisy and ground truth pairs for training.

## 2. Method

### 2.1. Dataset

We capture data with a transient lighting-enabled (Velten et al., 2020), clinical wide-field FGS imaging system (OnLume Surgical, Madison, WI). OnLume's system uses two camera sensors for both the reference and fluorescent cameras. In a surgical training model (Albano et al., 2021), we inject the near-infrared fluorescent agent, ICG, via syringe into the femoral artery of four chicken thighs to simulate vascular surgery. We prepared varying doses of ICG up to the clinical guidelines of 2.5 mg/mL to generate fluorescent videos with visual contrast and low noise that can be treated as ground truth and simulate much lower fluorescence. In future work, we would like to detect markers that have much fewer photons than our captured fluorescent videos. We capture about 100 minutes of simulated surgical footage with a variety of motion such as cutting, pulling, squeezing, injecting, and working with surgical tools. The 100 minutes of footage is broken up into 590 100-frame long videos. The videos are captured at 15 frames per second at a resolution of $768 \times 1024$; however, we downsample the resolution to $192 \times 256$ to speed up training times in our experiments.

To simulate our noisy LLV, we scale the fluorescent frames between $[\phi_{\text{back}}, \phi_{\text{sig}} + \phi_{\text{back}}]$ photons per frame, where $\phi_{\text{back}}$ is the number of background photons and $\phi_{\text{sig}}$ is the maximum number of signal photons in a pixel, and then we apply Poisson noise to the scaled frames. We use $\phi_{\text{back}} = 10$ photons accounting for ambient light and sensor dark counts. We use 3 different signal photon levels, $\phi_{\text{sig}} = [1, 5, 20]$, to represent a range of fluorescent strengths. We measure the noise level with the signal-to-background (SBR) ratio, SBR $= \frac{\phi_{\text{sig}}}{\phi_{\text{back}}}$. Giving us three SBR levels, SBR$= [0.1, 0.5, 2]$ or standard deviations of $\sigma = [826, 180, 57]$ on an 8-bit image, see Appendix E.

### 2.2. Denoising Algorithm

Our goal is to denoise the LLV with the help of a co-registered noise-free RV. For our noise model, we consider the case where the LLV was captured with a photon-limited camera such as a SPAD and is dominated by Poisson noise (Hasinoff, 2014). While a pure Poisson noise model is a good approximation for single photon cameras, commercial CMOS cameras may introduce additional noise terms but next-generation CMOS cameras (e.g. Hamamatsu qCMOS) have low enough noise to measure single photons. These new cameras are shot noise limited and generally have a Poisson dark current or dark count rate which can be modeled with a constant photon background rate. We anticipate next-generation cameras to be used in FGS, so we use a Poisson noise model with a constant background to simulate these cameras. We note that a different camera-specific noise model would be straightforward to implement with our model.

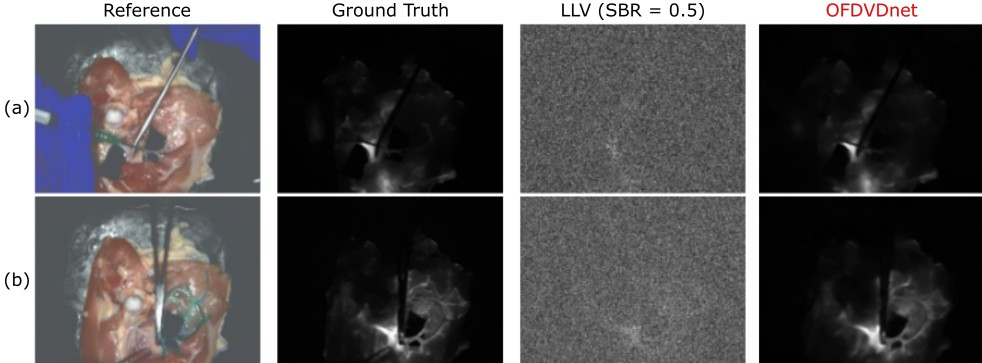

| Reference | Ground Truth | LLV (SBR = 0.5) | OFDVDnet |
|---|---|---|---|

**Figure 3: Injection of ICG:** In this figure, we show the RV frame, ground truth fluorescence frame, noisy LLV frame, and our proposed denoised result of a scene before and after ICG injection. (a) At the start of ICG injection, there is a small amount of innate fluorescence near the injection site. (b) After injection, the vascular structure becomes visible in the fluorescence channel. OFDVDnet is able to reconstruct both scenes well, maintaining most of the important vascular structure in (b). Note that at high concentrations the dye appears green, but this color does not necessarily indicate the near infrared fluorescence from the ICG for example see the tip of the syringe in (a).

At time $t$, let $\tilde{F}_t$ and $W_t$ denote the captured LLV frame and RV frame, respectively. Let, $\tilde{F}_t = Poiss(F_t)$ where $F_t$ represents the ground truth LLV frame, and $Poiss$ represents Poisson sampling. Because we consider very noisy cases it is difficult to denoise $\tilde{F}_t$ alone, so we use the RV as a guide.

We use a two-step denoising method, first our optical flow based denoising exploits the alignment of the LLV and RV to find motion within the scene from the RV and apply it to the LLV. Once the LLV is aligned, frames can be merged to reduce noise to create the optical flow denoised video (OFDV). We warp the LLV both forward and backward in time aggregating information from all frames. However, if a video is needed to be displayed with little latency, then the OFDV could be warped only forward in time resulting in lower latency at the cost of quality. In our second step, a CNN takes the OFDV and RV frames as input to further denoise. This two-step approach uses frames from distant points in time to denoise a single frame without increasing GPU memory use.

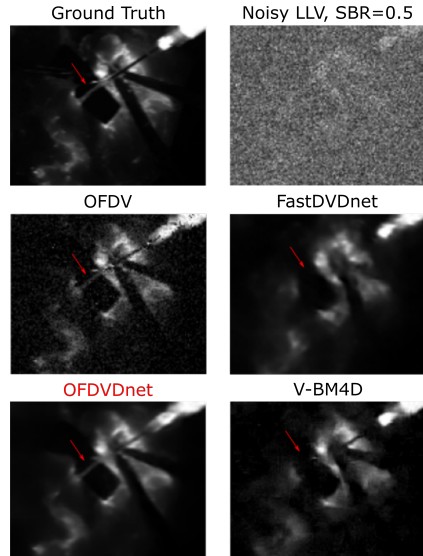

| Ground Truth | Noisy LLV, SBR=0.5 |
|---|---|
| OFDV | FastDVDnet |
| OFDVDnet | V-BM4D |

**Figure 4: Vessel Detail:** OFDVDnet correctly reconstructs a small vessel (red arrow) while comparison methods remove it entirely.

**Optical Flow Based Denoising:** One technique to denoise $\tilde{F}_t$ is to average $2T$ frames around time $t$, $\tilde{F}_{t-T}...\tilde{F}_{t+T}$. Higher $T$ will reduce noise but also increase motion blur. Therefore, we would like to spatially align the frames together before averaging. We use the RV to find optical flow and flow failure masks, which are then applied to the LLV to create the OFDV. The method is shown in Fig 2.

First, we use the RV frames to find optical flow warp maps, $A_{t\rightarrow t+1}$ that warp a frame at time $t$ to time $t+1$. $A_{t\rightarrow t+1}$ can be found using any optical flow method. We found that Gunnar-Farneback's (Farnebäck, 2003) optical flow algorithm was sufficient for our problem and, on our hardware, ran faster than the CNN based MaskFlowNet (Zhao et al., 2020). We use $A_{t\rightarrow t+1}$ to average the LLV frames along motion trajectories with a running sum strategy. Let $D_t^+$ be the running sum at time $t$, then,

$$D_t^+ = A_{t-1\rightarrow t}(D_{t-1}^+) + \tilde{F}_t \tag{1}$$

Ideally, we could estimate a frame at time t as $\frac{D_t^+}{t}$. However, optical flow can fail in a variety of circumstances such as occlusions. To deal with flow failures, for each frame we detect pixels with flow errors and reset the summation for those pixels. Then due to the resets each pixel in $D_t^+$ may represent a sum over a different period of time, so we also record the length of time since the last reset in each pixel given by $N_t^+$. Therefore, the new LLV estimate at time $t$ is given by $\frac{D_t^+}{N_t^+}$, which represents an average in each pixel since the last optical flow failure. Let $M_t$ be a binary mask which has a pixel value of 0 if $A_{t-1\rightarrow t}$ fails to warp correctly, then Eq. 1 becomes,

$$D_t^+ = M_t \odot A_{t-1\rightarrow t}(D_{t-1}^+) + \tilde{F}_t \tag{2}$$

$$N_t^+ = M_t \odot A_{t-1\rightarrow t}(N_{t-1}^+) + \mathbb{1} \tag{3}$$

where $\odot$ represents pixel-wise multiplication and $\mathbb{1}$ represents an image of all ones. We generate $M_t$ by detecting optical flow failures by comparing intensity values of successive reference frames; we warp $W_{t-1}$ and compare it to $W_t$ by,

$$M_t = \left| 1 - \frac{A_{t-1\rightarrow t}(W_{t-1})}{W_t} \right| < \tau \tag{4}$$

where $\tau$ is a threshold value, we use $\tau = 0.07$. We then compute, $\widehat{F}_t^{flow+}$,

$$\widehat{F}_t^{flow+} = \frac{D_t^+}{N_t^+} \tag{5}$$

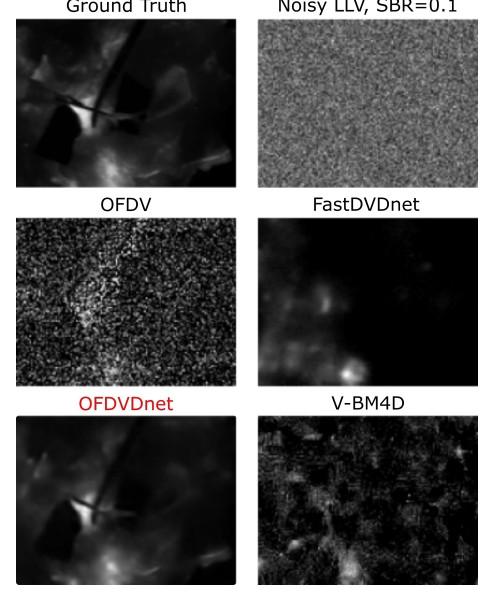

**Figure 5: High Noise:** This scene has no occlusions so OFDV averages over all frames giving good results where comparisons fail.

$\widehat{F}_t^{flow+}$ is the forward OFDV, we combine $\widehat{F}_t^{flow+}$ with the backward OFDV $\widehat{F}_t^{flow-}$ to create out final OFDV. The backward OFDV is calculated by running the same process reversed in time. Similar to the forward OFDV, we calculate $D_t^-$, and $N_t^-$ that represent the running sums on a time-reversed video. We can then create the final OFDV, $\widehat{F}_t^{flow}$, by combining the forward and backward estimations as follows,

$$D_t = D_t^+ + D_t^- - \tilde{F}_t \quad (6) \qquad N_t = N_t^+ + N_t^- - \mathbb{1} \quad (7) \qquad \widehat{F}_t^{flow} = \frac{D_t}{N_t} \quad (8)$$

where each pixel in $D_t$ and $N_t$ represents the sum and number of aligned pixels between two optical flow failures, respectively. Note that to find $D_t$ and $N_t$ we need to subtract out

**Table 1:** Comparison of PSNR/SSIM/FSIM (higher is better) for LLV frames denoised using OFDVDnet, OFDV, and the comparison methods at 3 noise levels.

| SBR | OFDVDnet | OFDV | FastDVDnet | V-BM4D | Guided Filtering | Joint Bilateral |
|-----|----------|------|-----------|--------|------------------|-----------------|
| 0.1 | **29.3/.76/.88** | 10.8/.015/.20 | 24.3/.48/.83 | 19.7/.19/.52 | 16.4/.19/.69 | 15.8/.11/.59 |
| 0.5 | **34.0/.89/.93** | 21.5/.22/.52 | 30.8/.80/.88 | 29.9/.61/.86 | 28.1/.61/.86 | 26.3/.52/.81 |
| 2.0 | **36.9/.92/.95** | 30.8/.72/.82 | 35.7/.89/.93 | 36.7/.88/.92 | 33.7/.90/.92 | 31.5/.85/.90 |

what is contained in both forward and backward OFDVs. A pixel value in the OFDV is the average value of an aligned LLV between optical flow failures. For example, if optical flow is correctly found for all frames, the OFDV will average along motion trajectories over the entire video, lowering noise and avoiding motion blur. Whereas if there is an occlusion the OFDV will avoid motion blur and only average pixel values between successive occlusion events. This process leads to the OFDV having high temporal consistency from frames being correlated and spatially varying noise levels from different averaging lengths per pixel.

**Neural Network:** In order to further remove the remaining noise and any warping artifacts in $\widehat{F}_t^{flow}$, we use a CNN denoiser based on FastDVDnet (Tassano et al., 2020) which takes five noisy frames to denoise the middle frame. We provide the CNN with five consecutive OFDV frames $\widehat{F}_{[t-2:t+2]}^{flow}$, averaging time maps $N_{[t-2:t+2]}$, and RV frames $W_{[t-2:t+2]}$ as input. We train the CNN to reconstruct the middle ground truth frame $F_t$ using 1000 training pairs over 100 videos with mean square error (MSE), see Appendix C,D for details.

The four neighboring OFDV frames of $\widehat{F}_t^{flow}$, $\widehat{F}_{\{t-2,t-1,t+1,t+2\}}^{flow}$, provide the CNN with additional information on the center frame and reduce flickering. The averaging time maps $N_{[t-2:t+2]}$ act as noise maps to indicate per-pixel noise levels because OFDV pixels have averaged signal over a varying number of frames and thus different noise characteristics. RV frames $W_{[t-2:t+2]}$ let the CNN exploit structural similarities between the RV and OFDV.

## 3. Results

Our testing set consists of the middle 96 frames from 100 videos. We compare our results to video denoisers only given the LLV frames, the CNN FastDVDnet (Tassano et al., 2020) re-trained on our data, and V-BM4D (Maggioni et al., 2012), a popular block matching and filtering technique. We also compare our technique to two image denoising techniques that use a RV frame to assist in denoising a LLV frame, guided filtering (He et al., 2012), and joint bilateral filtering (Gastal and Oliveira, 2011). In Fig 3, we show an example scene before and after ICG injection into the vessels in a chicken thigh. Before injecting ICG, Fig 3(a), fluorescence is only seen near the injection site and in areas of the chicken thigh that are either auto-fluorescent or fluorescent due to the chicken treatment process. As the dye injects into the femoral artery, it perfuses and smaller vascular branches begin fluorescing. OFDVDnet is able to reconstruct most of the details of the vascular system with only slight blurring.

OFDVDnet performs well in high noise when the OFDV can average over a large number of frames allowing use of information in the entire video to create the denoised output. Fig. 5 shows an example scene (SBR= 0.1) with no occlusion events and only small motion that can be easily taken care of by the OFDV. OFDVDnet reconstructs most of the fluorescent structure correctly, but struggles to reconstruct the moving syringe fully due to large movement. Both FastDVDnet and V-BM4D fail in this high-noise example.

We found that OFDVDnet could better reconstruct small vessels, one example is shown in Fig. 4. In this example, the OFDV contains the blood vessel that has been injected with the marker (red arrow) so the denoising network has the information needed to properly reconstruct this important feature. Whereas the vessel is lost in the noisy LLV frames so the comparison methods completely remove this vessel while showing a reasonable image. This failure mode is arguably worse than the un-processed noisy image since it suggests to the surgeon that the image is accurate but in fact conceals the vein.

By leveraging the correlation between successive OFDV frames, OFDVDnet produced videos with very little flickering. When our network denoises successive frames in a video the inputs are very similar, so the output will have little room to flicker due to noise fluctuations. Video results for OFDVDnet and the comparison methods can be found in Appendix B.

For our image quality metrics (IQM) we use peak signal-to-noise ratio (PSNR), as well as structural similarity(Wang et al., 2004) (SSIM), and feature similarity(Zhang et al., 2011) (FSIM) which better correspond to human interpretation than PSNR. The IQA results are summarized in Table 1. OFDVDnet outperforms at all noise levels tested and OFDVDnet's IQMs drop slower with increasing noise when compared to other methods.

**Network Ablation Study:** We study the effects of the reference frames, averaging time maps, and OFDV on the performance of our network. We retrain the network for each ablation case at SBR= 0.5. The results are summarized in Table 2. First, we tested removing the averaging time maps or the RV frame inputs which both decrease the PSNR of the result. Next, we test using only the RV frames as input which obtains a PSNR of 24.13 indicating strong structural priors, such as visible veins, in the RV.

**Table 2:** Ablation study PSNRs.

| SBR=0.5 | PSNR |
|---|---|
| OFDVDnet | 34.04 |
| No RV Frames | 33.82 |
| No Averaging Time Maps | 33.30 |
| Only RV Frames | 24.13 |
| Switch OFDV with 5 LLV | 31.05 |
| Switch OFDV with 17 LLV | 32.74 |

Finally, we replaced the OFDV input with the corresponding LLV frames which resulted in worse PSNR and significant flickering artifacts. We further studied the case of increasing the number of input LLV frames to 17, which was the maximum allowable due to GPU memory constraints. 17 LLV frames produced better quality results compared to five LLV. However, 17 LLV frames resulted in flickering artifacts and a decrease in PSNR compared to using five OFDV frames while also requiring a substantial increase in the required GPU memory (3x), training time (18x), and evaluation time (4x).

## 4. Conclusions and Discussion

In this work, we demonstrated a guided video denoising method meant for applications in FGS that is able to leverage deep learning in a memory-efficient manner with an explicit align and merge step. We captured and evaluated our method on a new dual-camera FGS dataset. OFDVDnet makes use of three properties spatial, temporal, and RV correlations to the LLV while comparison methods only use two of the three. FastDVDnet (Tassano et al., 2020) and V-BM4D (Maggioni et al., 2012) use spatial and temporal properties to provide decent results without the need for the RV, but fall off quickly as the noise level increases showing the importance of the RV at high noise. Guided filtering (He et al., 2012), and joint bilateral filtering (Gastal and Oliveira, 2011) make use of the RV, but only spatially, and use no temporal information. These methods have the worst image quality metrics and

result in flickering in the final video at high noise levels but are computationally the least expensive. Finally, our intermediate OFDV only makes use of temporal information from the RV and its results show the importance of using spatial information to denoise.

While OFDVDnet is able to produce strong results, we identify three key areas that future work can improve upon. First, OFDVDnet relies on the time-consuming computation of optical flow between the RV frames which disallows real-time use. An efficient patch-based approach (Hasinoff et al., 2016) may be faster using the RV frames as a guide. Second, OFDVDnet struggles when strong motion or occlusion disrupts the averaging of the OFDV leading to higher noise results. Better motion tracking that deals with these cases will be useful in denoising more challenging scenarios. One possible place to improve is in the detection of optical flow failures (Eq. 2.2); for example, the current method relies on relative intensity which will falsely detect a failure under changing lighting conditions this could be improved by using a different failure detection method such as normalized cross-correlation. Finally, our dataset is limited to simulated chicken thigh data, so it is unclear how learned priors will translate to other applications such as oncology. While our dataset is useful for evaluation of new algorithms, new application-specific datasets will be required for learning-based approaches. We hope our dataset and method can be used in further algorithm development for medical imaging applications that require higher signal under scene motion.

**Acknowledgments** This material is based upon work supported by the National Science Foundation Graduate Research Fellowship Program under Grant No. DGE-1747503 and CAREER Grant No. 1846884. Any opinions, findings, and conclusions or recommendations expressed in this material are those of the author(s) and do not necessarily reflect the views of the National Science Foundation. Support was also provided by the Graduate School and the Office of the Vice Chancellor for Research and Graduate Education at the University of Wisconsin-Madison with funding from the Wisconsin Alumni Research Foundation. This project was also supported in part by grant 1UL1TR002373 to UW ICTR from NIH/NCATS

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

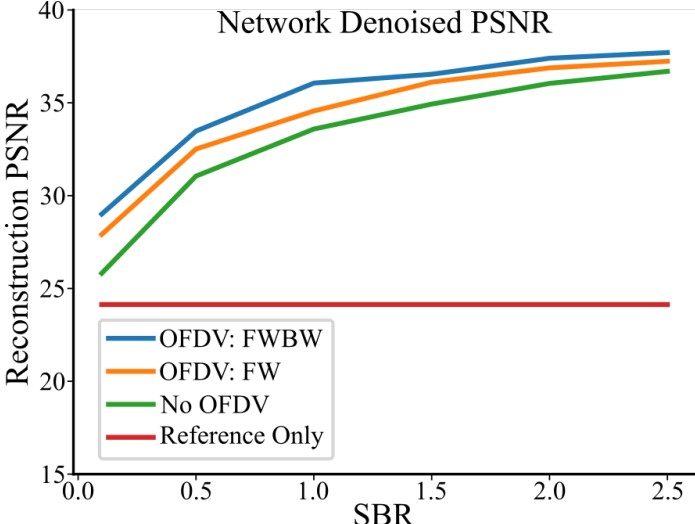

**Figure 6: Reconstruction PSNR at different SBR:** This plot shows our neural network reconstruction PSNR for a variety of SBR levels and different inputs into the network.

## Appendix A. Additional Ablation Studies

**Warp input frames:** Explicit motion compensation could benefit video denoising quality (Tassano et al., 2019). We implemented explicit motion compensation on the input data to our CNN by warping four of the five frames in $\widehat{F}^{flow}_{[t-2:t+2]}$, $N_{[t-2:t+2]}$, $W_{[t-2:t+2]}$ to align with their respective middle frames: $\widehat{F}^{flow}_t$, $N_t$, and $W_t$ using optical flow calculated from the reference frames. Warping did not improve video denoising quality for our case. This results in a PSNR of 33.92.

**OFDV forward vs. forward backward** OFDV forward(OFDV:FW) warps video frames forward in time, utilizing only information before the current frame in time space. OFDV:FWBW warps video frames both forward and backward in time, utilizing information from both before and after the current frame in time space.We compare the OFDV:FWBW to OFDV:FW by changing the inputs into the neural network at a variety of SBR levels. The results are summarized in Fig. 6 along with PSNR curves for the only reference frames and replacing the OFDV with the LLV. We find the OFDV:FWBW obtains the best results due to its ability to use many frames from both the future and the past. Using the OFDV:FW achieves strong PSNR; however, more flickering is observed in the denoised results when compared to the OFDV:FWBW. This is likely due to stronger input correlations and lower noise in the OFDV:FWBW.

**OFDV Ablation** Next, we examine how the tunable parameters in our OFDV construction effect the performance of the OFDV construction. Because training the network from scratch takes about 1 week on our hardware we instead choose to run a study on the OFDV PSNR performance. Our OFDV relies on the computation of optical flow based on OpenCV's implementation (Bradski, 2000) Gunnar-Farneback's algorithm (Farnebäck, 2003) which has 6 tunable parameters. We also have a seventh tunable parameter in our detection of optical flow failures, $\tau$, that controls the sensitivity of our optical flow failure detection where higher values lead to fewer failure detection events. We compute the PSNR of the OFDV on one-third of the training set for a range of values of all 7 of these parameters. For each parameter, we calculate the PSNR for a specific value as the maximum PSNR achieved with respect to the other 6 parameters. We found that the 6 optical flow parameters had very little impact on the final OFDV performance changing by less than 0.25dB over the range of tested values (ramge chosen based on suggestions from OpenCV's documentation). We found $\tau$

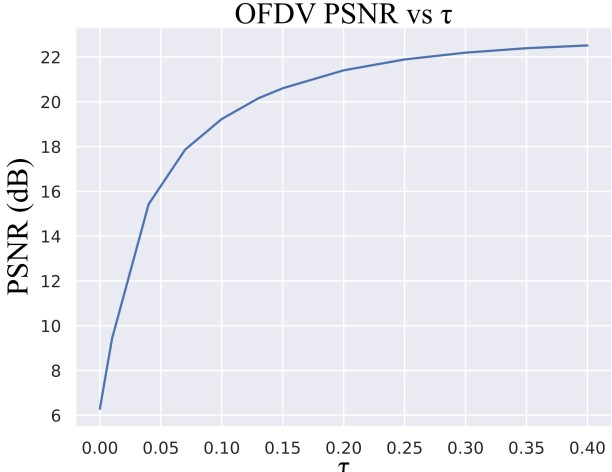

**Figure 7: OFTV PSNR vs $\tau$:** This plot shows our OFDV reconstruction PSNR with respect to changing the $\tau$ parameter.

has a much larger impact on OFDV PSNR and find that larger $\tau$ is useful before a limit is reached, these results are shown in Fig 7.

## Appendix B. Video Results

We have included 3 videos from the test set at each noise level. In each video we include the reference video, ground truth LLV, noisy LLV, the OFDV, our result, FastDVDNet's(Tassano et al., 2020) result, and V-BM4D's(Maggioni et al., 2012) result. A brief description of each included video:

- Video 323: There is slight movement in this video from tool use.

- Video 383: In this video there is moderate movement due to pulling tissue.

- Video 499: The camera is bumped in this video leading to large motion in the scene.

We also include a video of denoised results from our ablation study. The result includes 25 test videos from the neural network denoised output with different input configuration. We use the following different inputs: OFDV:FWBW, OFDV:FW, 5 noisy LLV frames, 17 noisy LLV frames, and only reference frames. These results are at $SBR = 0.5$.

We share our video results on our YouTube channel: https://www.youtube.com/@fgs_denoising/playlists. For best viewing quality, we recommend enabling "fullscreen" by clicking the bottom-right button in the YouTube video player when viewing the videos. See the "About" tab in the channel for a short description on navigating the channel.

The original videos can also be downloaded from our Google drive:
https://drive.google.com/drive/folders/1QarTJx4h7TiVlR2zekRwcmqoV9fRL8Je?usp=sharing.
The included .txt file provides a short description on how to navigate the folders.

## Appendix C. Neural Network Architecture

We use the same general network architecture as FastDVDNet (Tassano et al., 2020) for our video denoising network. Suppl. Fig. 8 shows a diagram of the architecture. When denoising an OFDV frames $\widehat{F}_t^{flow}$, the network takes four of its neighboring OFDV frames $\widehat{F}_{\{t-2,t-1,t+1,t+2\}}^{flow}$ along with

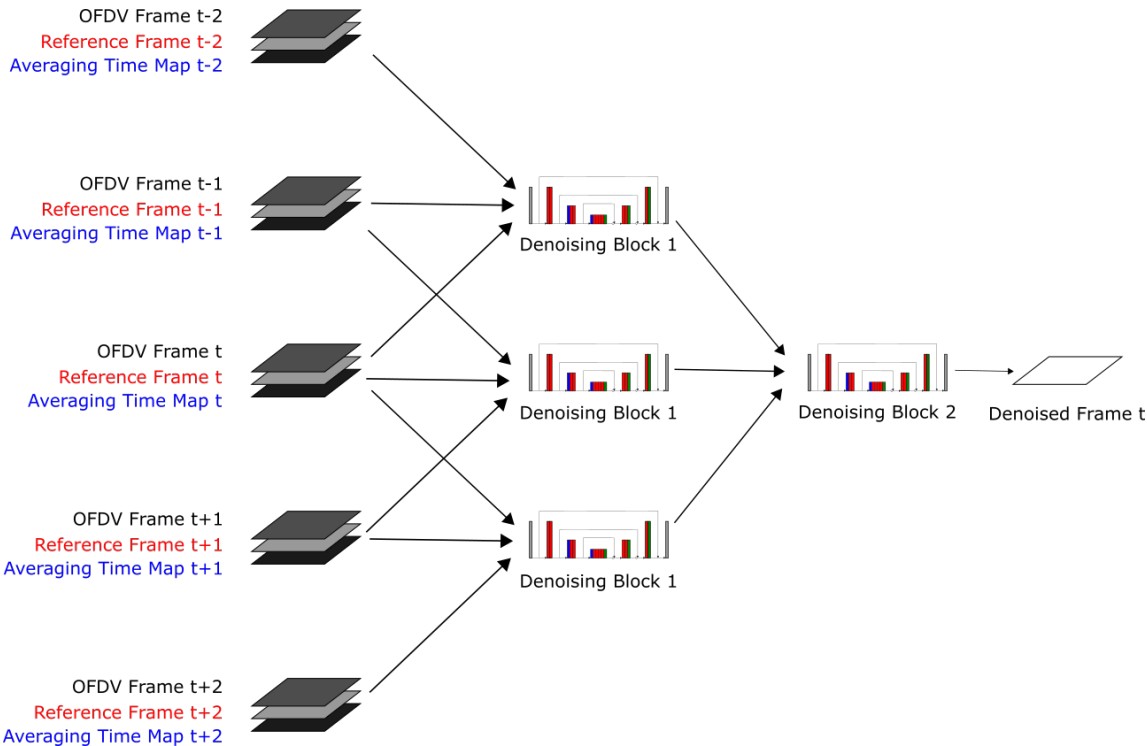

**Figure 8: Diagram of the video denoising network**

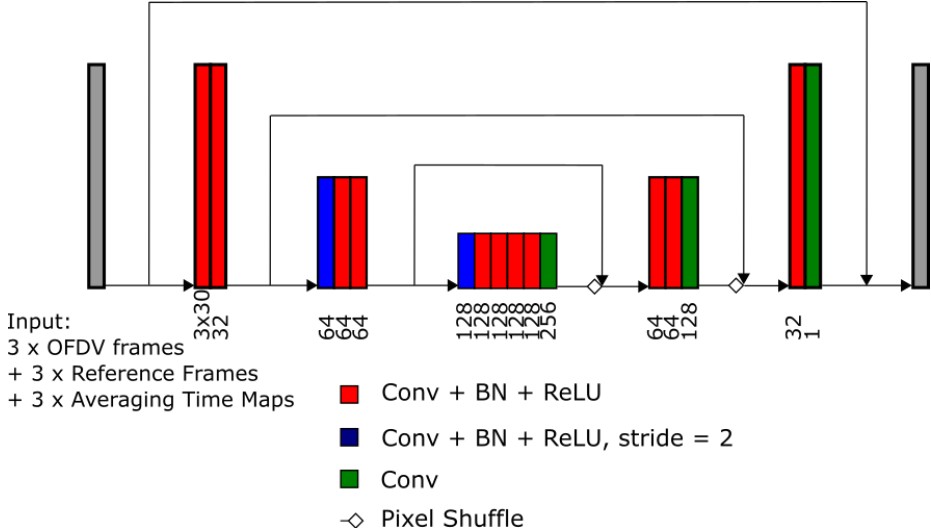

**Figure 9: Architecture of the denoising block**

their corresponding reference frames $W_{[t-2:t+2]}$ and averaging time maps $N_{[t-2:t+2]}$ as inputs. During the forward pass, adjacent OFDV frames along with their corresponding reference frames and averaging time maps are passed into denoising blocks in groups of three. A diagram of the denoising block is shown in Fig. 9.

## Appendix D. Training Details

Our CNN is trained on the output frames from the optical flow denoising algorithm or OFDV frames. The training dataset consists of input/ground-truth frame pairs $P_t^k$:

$$P_t^k = \left( (\widehat{F}_{[t-2:t+2]}^{flow,k}, W_{[t-2:t+2]}^k, N_{[t-2:t+2]}^k), F_t^k \right)$$

where $k \in \{1, 2, \cdots, n_{\text{tot}}\}$.

For a given noise level, the training dataset contains a total number of $n_{tot} = 1000$ input/ground-truth pairs. We sample 1000 ground-truth fluorescent frames from the first 250 out of 590 ground-truth LLVs, and fill in their corresponding OFDV frames $\widehat{F}_{[t-2:t+2]}^{flow,k}$, reference frames $W_{[t-2:t+2]}^k$, and averaging time maps $N_{[t-2:t+2]}^k$.

We use Mean Squared Error (MSE) as our loss function:

$$L(\theta) = \frac{1}{n_{\text{tot}}} \sum_{k=1}^{n_{\text{tot}}} \|\widehat{F}_t^{net,k} - G_t^k\|^2 \tag{9}$$

where $\theta$ is the set of learnable parameters; $\widehat{F}_t^{net,k}$ is the output of the CNN.

ADAM optimizer(Kingma and Ba, 2015) is used to minimize the loss function with all its parameters set to default values and the initial leaning rate set to $10^{-3}$. The network is trained for 100 epochs with a batch size of 8. Our CNN is trained separately for three different noise levels, SBR $= [0.1, 0.5, 2]$.

**Retraining FastDVDNet:** When comparing our method to FastDVDNet (Tassano et al., 2020) we found it was necessary to retrain FastDVDNet on our dataset and noise levels in order for FastDVDNet to produce reasonable results. We feed FastDVDNet the raw LLV frames and an estimated noise map. We estimate a constant noise map for each frame by using the average number of photon counts captured at each frame to estimate the noise standard deviation (see Appendix E). We train FastDVDNet using the same training procedure as our method.

## Appendix E. Finding Equivalent Gaussian Standard Deviation

Before we generate Poisson noise we scale our images between $[\phi_{\text{back}}, \phi_{\text{sig}}]$ then we add Poisson noise and re-scale the noisy image between [0,255]. For comparison to methods using the standard deviation of additive white Gaussian noise, the comparable standard deviation of our Poisson corrupted images scaled between [0,255] is given by,

$$\sigma = \sqrt{\phi} \frac{255}{\phi_{\text{sig}}} \tag{10}$$

where $\phi \in [\phi_{\text{back}}, \phi_{\text{sig}}]$ is the expected numbers of photons for a given pixel. Although, Poisson noise is signal dependent we can get an estimate of the noise in a scene by using $\phi = \frac{1}{2}(\phi_{\text{sig}} + 2\phi_{\text{back}})$, which gives us a standard deviation of $\sigma = [826, 180, 57]$ for our three noise levels of SBR$= [0.1, 0.5, 2]$.

## Appendix F. Optical Flow Failures

Figure 10 shows five consecutive LLV, RV, OFDV denoised, OFDVDnet denoised and ground truth frames. The red box circles out a region with optical flow failures due to rapid occluder (a pair of tweezers) movements in that area. The rapid movement of occluder causes optical flow to fail repeatedly in short time intervals for those regions around the occluder, leading to less pixels being averaged over time and more noise remaining in those pixels. As shown in Figure 10, for each of the five OFDV frames, the area inside the red box has significantly more noise compared to areas outside the box. Because the OFDV frames are more noisy in areas with rapid occluder movements,

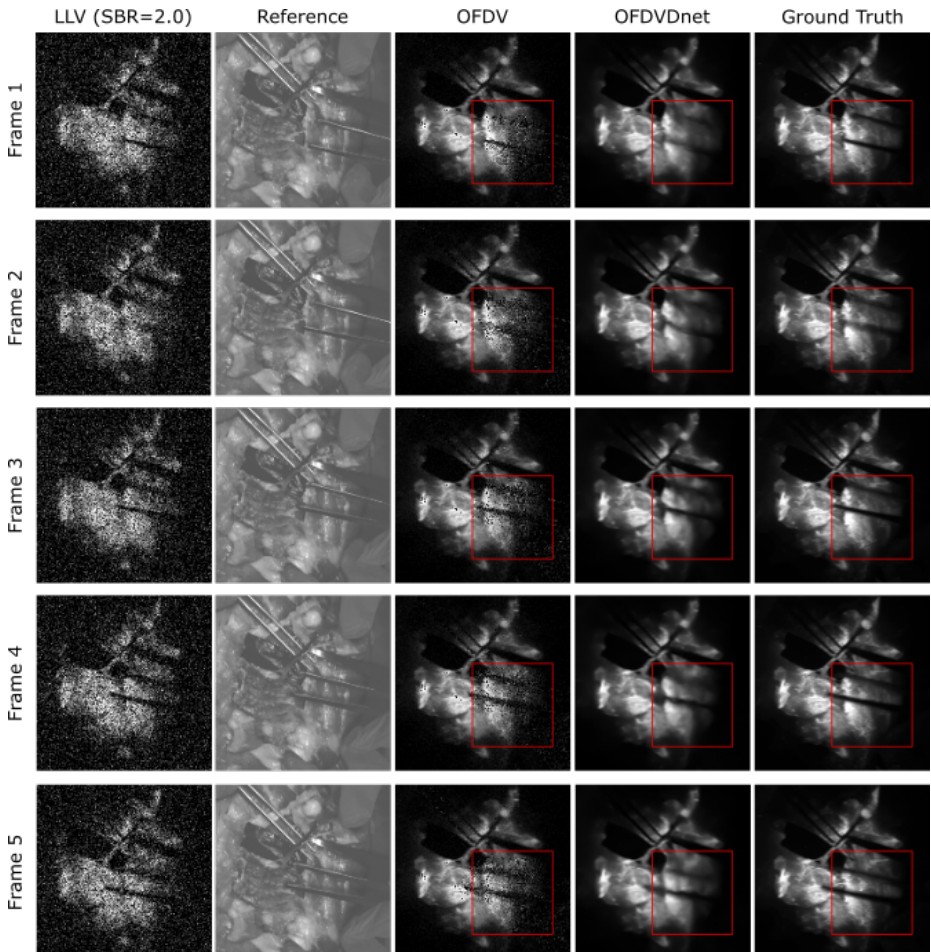

Figure 10: Example case with optical flow failures

the OFDVDnet also tend to blur out more details in those areas compared to other areas in the video frames during the neural network denoising step.

