# OpenReview forum: "OFDVDnet: A Sensor Fusion Approach for Video Denoising in Fluorescence-Guided Surgery"
_MIDL.io/2023/Conference — MIDL 2023 Poster_

### Official Review · Reviewer_mpus · 2023-02-01

**Confidence:** 3
**Preliminary Rating:** 5
**Recommendation:** Poster

**Summary:**

The authors present OFDVDnet, a hybrid method that combines the align and merge technique and deep learning technique for FGS denoising.
In the first stage,  optical flow denoised video (OFDV) is estimated which is a compressed representation of many different frames.

The ODFV and the RV frames are fed into a video-denoising neural network inspired by FastDVDnet to create a final denoised output.

They have used the deep learning model to consider scenarios of extreme noise that are not handled by the align and merge technique.  OFDVDnet makes use of three properties spatial, temporal, and RV correlations to the LLV while comparison methods only use two of the three.


**Strengths:**

The authors have considered an important problem of LLV denoising in real time, FGS, for accurate intraoperative decision-making.  The authors clearly explain the method of OFDVDnet and compare it to other existing
methods in terms of performance and computational efficiency.

The flow of the introduction with enough background content, the limitation of existing methods, and the proposed method is really good.

The methodology captures details such as flow failures, a record of the number of aligned pixels between two optical flow failures, temporal correlation, and spatially varying noise levels.

The details taken from the previous two and next two frames help to recover fine vessel details well for the CNN denoiser.

The authors conclude by stating that OFDVDnet can produce strong results and also
identifies three key areas for future improvement in the method such as making the
method faster and better at handling motion and occlusion.

The dataset preparation of 100 minutes of footage is impressive work.


**Weaknesses:**

On page 5, the authors introduce warp maps from time t to t+1 but suddenly equation 1 considers warp maps from t-1 to t. The authors can choose consistent notations to explain the methodology.

The significance of equation 4 is not explained.

Figure 7, does not highlight notations of inputs OFDV frame, reference frame, and averaging time map.

Note that to find Dt and Nt we need to subtract out what is contained in both forward and backward OFDV ---> the authors should rephrase this sentence. It is not clear, what "what" means.

Does the methodology capture details such as flow failures but the results (quantitative and qualitative) of OFDVDnet for this case are missing? For instance, partial occlusions can be a case.

Other evaluation metrics along with PSNR like structural similarity index (SSIM),
Recognition accuracy (RA) or Pixel Error Rate (PERR) can also be considered.


**Deanonymize Review:**

no

**Detailed Comments:**

The authors talk about the overview of OFDVDnet in Fig.1. In the figure, they mention the mask on Page 3, but the mask is absent in Fig. 1. It is present only in the detailed view in Fig. 2. To improve readability, the authors can either add a mask block in Fig. 1 or add a reference to Fig. 2 when introducing occlusion mask.


The notation  (n=4) is not defined.

we will combine on page 6 after equation 5 ----> The authors must stick to tenses. -----> We combine


 (see E) page 14 is not clear.


 We retain the network for each case at SBR= 0.5. --------> this sentence needs to be clarified.

 The spelling for Reference is wrong in Figure 7.


**Paper Type:**

validation/application paper

**Questions To Address In The Rebuttal:**

Why the qualitative results for occlusion / partial occlusion cases are not discussed? Occlusion is a critical case where optical flow fails right?

Is Figure 4, does the loss of vessel detail an occlusion case or a case that showcases the recovery of fine details?

---

### Official Review · Reviewer_xQi5 · 2023-02-02

**Confidence:** 3
**Preliminary Rating:** 4
**Recommendation:** Poster

**Summary:**

This manuscript proposes OFDVDnet, a learning-based framework for denoising fluorescent video. The approach attempts to exploit spatial/temporal/reference information for better visualization. Reference videos are used to compute optical flow for inter-frame warping. Fluorescent videos use optical flow information for temporal alignment. The aligned frames are aggregated and passed to a neural network for denoised output. The experiments are conducted on an in-house dataset. The results show improvements over various baselines.

**Strengths:**

The paper is well-written with clear details and illustrations. The methodology is sound with convincing motivation. The in-house dataset is skillfully collected. The experiment results are sound, demonstrating the advantages of the approach.

**Weaknesses:**

- Minor ablations missing -- lack of ablation on sensitivity to thresholds $\tau$
- Minor suggestion – replacing relative intensity difference with NCC may be more robust, yet not much computation overhead.


**Deanonymize Review:**

no

**Detailed Comments:**

Minor ablations missing -- lack of ablation on sensitivity to thresholds $\tau$ used in Eqn 4
- How sensitive the approach can be w.r.t $\tau$?

A followed-up suggestion
- Intensity variation is not accounted for when using relative intensity difference (Eqn 4). Have you tried normalized cross-correlation? Are there performance improvements?

Misc
- How well is the reference video and fluorescent videos aligned?


**Paper Type:**

methodological development

**Questions To Address In The Rebuttal:**

Overall, the paper quality is good with a clear description. If the authors can clarify the details I mentioned, that will be great. However, these missing details are not the motivation for rejection.

---

### Official Review · Reviewer_XKkf · 2023-02-09

**Confidence:** 3
**Preliminary Rating:** 4
**Recommendation:** Poster

**Summary:**

This paper presents a video denoising method for fluorescence image-guided surgery. The core idea is to use optical flow and an occlusion mask between successive frames to create a denoised motion-compensated estimate. The goal is to recover a denoised video from a noisy low-light video. A deep learning model is used to further remove the remaining noise and any warping artifacts.


**Strengths:**

This application is on an interesting topic, fluorescence image-guided surgery. The success of denoising will improve the quality of the surgery.

The motivation is well illustrated in Introduction.

OFDV is it is a compressed representation of many different frames that can be GPU memory efficient.

The method outperforms the listed baseline method.

The figures are informative.




**Weaknesses:**

Lack of empirical study. The training and testing data are from simulation rather than empirical studies.

The paper uses a number of background photons and the maximum number of signal photons in a pixel and applies Poisson noise to the scaled frames. However, it is not clear why that is a good enough simulation strategy and real noise distribution in real fluorescence-guided surgery.

To deal with optical flow failures, the strategy in Fig 2. has been developed. However, it is not clear what is the failure rate (how frequent) and how bad the failures can be.

The testing set consists of the middle 96 frames from 100 videos. Since the frames have a strong temporal correlation. This strategy might not be rigorous enough. This can be alleviated by using independent testing videos

The performance gain of OFDVDnet (PSNR 34.04) is only marginally better than No RV Frames (PSNR 33.82) and No Averaging Time Maps (PSNR 33.30). Could this improvement make the clinical practice fundamentally better?





**Deanonymize Review:**

no

**Detailed Comments:**

What is the spatial and temporal resolution of the videos?

The way that this paper is used to deal with the failure cases is empirical. From the methodological description, it is not guaranteed that this method would work.

Why we don't directly use the noisy videos as inputs, while the ground truth as outputs is direct learning?






**Paper Type:**

methodological development

**Questions To Address In The Rebuttal:**

Please address/answer the concerns in the Weaknesses section and the Detailed Comments section. My mind might be changed if I got a persuasive rebuttal from the author team ( but it not always guaranteed)

---

### Meta-Review · Area_Chair_und5 · 2023-02-24

**Recommendation:** Accept (Poster)
**Confidence:** 5

**Metareview:**

There is concensus among the reviewers that the manuscript is well written, presents an interesting approach, and convincing results.

Major concerns pertaining to the validity of simulation and data splits were addressed successfully. It appears that the only concern preventing an even more positive assessment would be the demonstration of the method on real data, which while planned for the future, is not demonstrated here.

Considering that the major shortcomings have been addressed and reviewers are satisfied with the revisions, this manuscript can be accepted.